# CoMNet: Cortical Modules Are Powerful

## Abstract

Existing CNN architectures may achieve efficiency in either one or two dimensions: FLOPs, depth, accuracy, representation power, latency but not in all. In this work, we present a pragmatically designed novel CNN architecture "CoMNet" which offers multi-dimensional efficiency at once such as: simple yet accurate, lower latency and FLOPs, high representation power in limited parameters, low memory consumption, negligible branching, smaller depths, and only a few design hyperparameters. The key to achieve the multi-dimensional efficiency is our use of biological underpinnings into CoMNet which is primarily the organization of cortical modules in the visual cortex. To realize CoMNet, a few concepts from well understood CNN designs are directly inherited such as residual learning. Our solid experimental evaluations demonstrate superiority of CoMNet over many state-of-the-art industry and academia dominant architectures such as ResNet, RepVGG etc. For instance, CoMNet supersedes ResNet-50 on ImageNet while being $50\%$ shallower, $22\%$ lesser parameters, $25\%$ lower FLOPs and latency, and in $16\%$ lesser training epochs. Code will be opensourced post the reviews.

## 1 Introduction

To date, a wide variety of CNN architectures exists: branchless Krizhevsky et al. (2012); Simonyan & Zisserman (2014), single branch (He et al., 2016), multi-branch and variable filter topology (Szegedy et al., 2015; 2016), feature resue (Huang et al., 2017), mobile series (Howard et al., 2017; Zhang et al., 2018). Despite their architectural diversity, they all have one characteristic in common; their primary design objective dimensions are limited. For instance, (Simonyan & Zisserman, 2014; Szegedy et al., 2015; He et al., 2016; Huang et al., 2017) seek to achieve higher accuracy regardless of network size and runtime, while mobile networks (Howard et al., 2017; Zhang et al., 2018) aim for fewer FLOPs at the cost of reduced representation power. Some neural architecture search based designs (Zoph et al., 2018; Tan & Le, 2019) focus on both higher accuracy and lower FLOPs but are heavily branched, discard latency, and runs slower on GPUs. More recently, (Ding et al., 2021) focus on accelerating (Simonyan & Zisserman, 2014) during inference phase while leaving training-phase unaddressed.

In contrast, there are many other dimensions which are left unaddressed but are crucial in real-time applications, such as autonomous driving, robotics etc. Foremost is latency or per-sample runtime which still remains ignored in CNNs in favour of FLOPs or throughput. Optimizing latency is crucial because real-time applications tend to process as few as one frame at a time but not large batches, and also, fewer FLOPs does not mean lower latency (Ding et al., 2021). Second, neural networks are replacing their traditional coutnerparts, posing a huge challenge to run multiple networks on a single device (Kumar et al., 2020), bringing in resource efficiency constraints. Third, rising complexity of the tasks (Kumar et al., 2020; Kumar & Behera, 2019) demands large networks to achieve satisfactory results, because mobile networks are insufficient due to their smaller representation power (Sec 2).

In the existing designs, the primary dimensions such as higher accuracy (He et al., 2016; Szegedy et al., 2016), fewer FLOPs (Sandler et al., 2018), and inference time FLOPs (Ding et al., 2021) are explored only individually. However, considering the above discussion, addressing multi-dimensional efficiency is current need of the time. Consequently, in this work, we aim to achieve multi-dimensional efficiency at once, while offering stronger trade-offs in some of them if efficiency in all dimensions is not feasible. To the best of our knowledge, achieving multi-dimensional efficiency has not been explored as it is a difficult task, mostly due to a high correlation among dimensions, which makes it possible to be better in one but worse in another.

Towards our objective, we propose a novel architecture "CoMNet", that can offer multi-dimensional benefits of lower architectural complexity, smaller depth, hardware-accelerators compatible, low memory consumption, low memory access costs on parallel computing hardware, low latency, parameter efficiency. CoMNet is primarily based on our translation of biological underpinnings of cortical modules (Mountcastle, 1997) which predominantly exist in a visual cortex. We particularly

refer to the structure of cortical modules in ventral stream (Tanaka, 1996) which is responsible for object recognition in mammalians. To realize our mere biological design inspirations, we inherit a few concepts from the well established CNN designs e.g. residual leaning (He et al., 2016). CoMNet mostly outperforms many key CNN designs, and also offers trade-offs but minimally. Although CoMNet has its design inspired from biological studies, we neither investigate nor assert that CoMNet is functionally similar to the visual cortex.

In a brief, key take-aways of the paper are: *First*, the notion of multidimensional efficiency in CNNs, *Second*, the notion of artificial cortical modules (`ACM`) which helps achieving high representation in fewer parameters, controlled parameter growth, increased computational density, *Third*, the concept of columnar organization (Mountcastle, 1997) which helps to achieve smaller depths, lower latency and FLOPs, faster convergence, *Fourth*, long range connections similar to pyramidal neurons (Mountcastle, 1997) which further improves the accuracy of CoMNet. We also provide a detailed ablations, and a minimal design space of CoMNet to help one choosing a suitable model. By using GradCAM (Selvaraju et al., 2017), we also show that CoMNet learns better data representations. Finally, we briefly discuss BrainScore (Schrimpf et al., 2020) and future prospects of CoMNet.

In the next section, we talk about the most relevant works. In Sec. 2, we brief our biological insights and their translation into the CoMNet architecture. In Sec. 5, we present a rigorous experimental analysis. Finally, in Sec. 6, we provide conclusions about the paper.

## 2 RELATED WORK

**Parameters and Representation Power:**   The earlier CNN designs (Krizhevsky et al., 2012; Simonyan & Zisserman, 2014; Szegedy et al., 2015; He et al., 2016) possess high representation power. In these designs, the deeper layers consist of a large number of channels e.g. 512 (Simonyan & Zisserman, 2014) to compensate for the reduction in resolution, leading to exponential growth in the parameters, and synaptic connections of a kernel in these layers. It becomes a predominant cause of overfitting (Simonyan & Zisserman, 2014) which is alleviated via dropout (Krizhevsky et al., 2012), but at the cost of increased training time. ResNet (He et al., 2016) avoids that by reducing and expanding the number of channels via $1 \times 1$ conv layers and placing them before and after $3 \times 3$ convolutions. Xie et al. (2017) restructures the residual unit of ResNet in form of groups, however, the issue of large depth, overall parameters still remains in picture. The mobile CNNs (Howard et al., 2017; Sandler et al., 2018; Zhang et al., 2018; Ma et al., 2018) on the other hand employ depthwise convolutions (Sifre & Mallat) to control the parameter growth and reduce FLOPs. However representation power decreases quickly. Moreover, depthwise convs are devoid of cross channel context which is crucial for better performance (Zhang et al., 2018). Therefore such convolutions are followed by a $1 \times 1$ conv to intertwine cross channel information.

**Depth:**   In the above networks, the use of $1 \times 1$ layers increases network depth rapidly. For example, two $1 \times 1$ for each $3 \times 3$ in (He et al., 2016; Sandler et al., 2018; Zhang et al., 2018; Tan & Le, 2019), while one in (Howard et al., 2017). Despite being beneficial, these layers constitute a significant amount of depth e.g. 66% in (He et al., 2016). Moreover, due to their pointwise nature, $1 \times 1$ convs do not contribute in the receptive field, which in contrast is governed by $3 \times 3$ layers.

**Branching:**   With time, the CNN architectures have grown from branchless (Krizhevsky et al., 2012; Simonyan & Zisserman, 2014) to single branch (He et al., 2016) to multibranch (Szegedy et al., 2016; Schneider et al., 2017). Neural architecture search has resulted in even heavily branched designs (Zoph et al., 2018; Tan & Le, 2019). Although branching helps in improving accuracy, it also raises memory access cost on parallel computing hardware (Ding et al., 2021) which directly impacts latency and memory consumption.

**Latency:**   Both Depth and high branching increase latency despite having fewer FLOPs. In CNNs, each layer require certain computing time and many such layers are linked serially. Output of one layer can not be computed until all outputs of the preceding layers are available, despite enough computing power remains. This dramatically increases the latency despite fewer calculations per layer. For example, 100 layers each with 1ms time result in 100ms of latency while a shallow network of 15 layers with 3ms per layer runtime will result in 45ms of latency. The best illustration of this phenomenon is (Tan & Le, 2019) which despite having fewer FLOPs, runs equivalent to a five times bigger network (He et al., 2016). More recently, (Ding et al., 2021) proposes structural reparameterization to accelerate (Simonyan & Zisserman, 2014) during inference phase, however, the training-phase network is still high in parameters, and branches even more than its predecessor (He et al., 2016), showing no improvement in training time (Table 4).

**Convergence:**   Reduced representation power in (Howard et al., 2017; Sandler et al., 2018; Zhang et al., 2018; Ma et al., 2018; Tan & Le, 2019) results in overly large training schedules on ImageNet, and reduced performance on other downstream tasks. For instance, (Howard et al., 2017) requires 200 epochs on ImageNet to perform similar to one variant of (Simonyan & Zisserman, 2014) which is trained only for 75 epochs, but performs poorly on object detection. A similar case (Tan & Le, 2019) which requires $4\times$ training schedule(400 epochs), and larger input resolutions in contrast to the earlier ones (He et al., 2016; Simonyan & Zisserman, 2014; Schneider et al., 2017) that are always trained for a smaller resolution $224 \times 224$ and in a range of $90 - 120$ epochs.

## 3   BIOLOGICAL VISUAL CORTEX

A biological visual cortex is fairly complex but consists of several interesting properties. We highlight the most relevant ones below. For more details, kindly refer to the supplementary material.

**Columnar Structure.**   Cortical modules are present all across a biological cortex (Mountcastle, 1997). Modules in the shallower layers are referred as ocular dominance columns which respond to simple stimuli such as edges, lines of different orientations (Hubel & Wiesel, 1963). Whereas modules in the deeper layers are a collection of neurons which respond to a complex stimuli such as face, monkey, human etc, by encoding stimuli information from different viewpoints (Tanaka, 1996). These modules are mainly present in inferotemporal cortex (IT) which is responsible for object detection, and recognition tasks (Figure 1a).

**Shared Input or Input Replication.**   Two cortical modules having different stimuli response can have a common input i.e. the input is replicated to feed the modules. This is crucial because at a given location in the visual field, there can be a monkey or a car etc. Hence, a module with highest similarity with the stimuli fires strongly, and signals to other parts of the cortex (Hubel & Wiesel, 1963).

**Limited Synaptic Connections.**   Typically, a cortical column contains a small number of neurons $(70 - 100)$ (Mountcastle, 1997) which results in smaller number of synaptic connections per column. It is advantageous as this collection of neurons can learn a simple stimuli.

**Massive Parallelization.**   Regardless of layers being shallower or deeper, a cortical module processes only a small region of retina. Multiples of such modules having similar stimuli response are replicated to span the whole retinal field. This organization facilitates massive parallelization.

**Lateral Connection Inhibition.**   Cortical modules do not communicate with each other except at the output (Tanaka, 1996), which is achieved via fewer pyramidal neurons having large number of long range connections. Such neurons fuse cross module information (Tanaka, 1996).

## 4   COMNET DESIGN

While designing CoMNet, we closely pay attention to the five most crucial objectives: latency, depth, branching, FLOPs, and parameters, which are sufficient to achieve multi-dimensional efficiencies of controlled parameter growth, high representation power in fewer parameters, increased computational density, minimal branching, as other objectives such as memory consumption, memory access cost depend on these. Next, we discuss our proposed biologically inspired designs, their advantage in the context of the above objectives, and their translation into CNN realm. At the end of this section, we intertwine all the individual designs to develop the fundamental computational unit of CoMNet which is used to build different instances of CoMNet.

### 4.1   ARTIFICIAL CORTICAL MODULES

Our first contribution is to develop IT like structure in CNNs. We achieve that by developing artificial cortical modules ACMs, and a mechanism to feed them similar to input replication in the visual cortex. We realize input replication by defining an operator, called CM_Feeder. This operator transforms a tensor $\in \mathbb{R}^{C \times H \times W}$ into another tensor $\in \mathbb{R}^{(M \times C) \times H \times W}$, where $M$ denotes the desired number of cortical modules. This operator consumes a tensor and outputs its $M$ identical replicas (Figure 1b).

Since a cortical module is essentially a group of neurons, we realize its artificial counterpart (ACM) via a $k \times k$ convolution layer, $k = 3$, having $N$ neurons, Figure 1c. Following, the limited synaptic connection property, $N$ is kept small. To serve as $M$ cortical modules, $M$ such convolutions are employed in parallel, each processing one of the replicas obtained from the CM_Feeder. We refer the collection of $M$ ACMs as a Collective Cortical Module (CCM). The above proposed concept of ACM helps achieving controlled parameter growth, high representation power in fewer parameters, and improved computational density as discussed below.

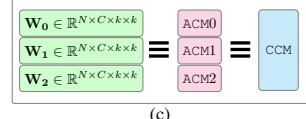

Figure 1: (a) Illustration of cortical modules in IT, (b) `CM_Feeder`, and (c) realization of `ACM`, `CCM`.

**Controlled Parameter Growth.** In the earlier designs, number of parameters is controlled by either increasing network width or depth which offers only a little control, and leads to exponential growth in parameters, especially in the deeper layers (Sec. 2). On the other hand, `ACMs` provide an absolute control over number of synaptic connections which precisely reflects the parameters count. This flexibility is crucial while scaling a CNN as per the needs.

**High Representation Power** We build a hypothesis that multiple neurons with lesser synaptic connections are better than a single neuron with large connections. For any stimuli, it requires certain amount of representation power or neurons to be learnt. In the existing CNNs, a kernel operates on a large number of channels, thus having large number of synaptic connections. During learning phase, it gets penalized for all the visual stimuli even if it is uncorrelated with them. This is also visible by weight pruning which suggests that post training, eliminating many channels/connections does not impact accuracy, indicating a wastage of synaptic connections.

CoMNet addresses the issue by prioritizing smaller neurons over larger ones by limiting the number of synaptic connections. In other words, we synthesize more number of neurons out of a larger one. As a result, CoMNet exhibits higher accuracy (Table 1) in lesser parameters without prolonging the training duration, in contrast to (Sandler et al., 2018; Tan & Le, 2019). It shows that CoMNet learns better representations in fewer parameters (Figure 3).

**Increased Parallelization and Computational density.** Naively, each `ACM` in a `CCM` is executed independently, however as all of the `ACMs` are operating parally, they can be computed efficiently by using NVIDIA's highly optimized `Batched-Matrix-Multiply` routines. To achieve that, we combine all the `ACMs` of a `CCM` layer into a single convolution layer having $M$ groups. This strategy packs computations of all `ACMs` into a single convolution, which leads to increased computational density, increased GPU utilization, and reduced memory access cost (Ding et al., 2021).

Although group convs are widely explored (Xie et al., 2017; Zhang et al., 2018), there exist two key differences. *First*, in this approach, input channels are divided into groups i.e. a slicing operation, whereas in CoMNet, inputs is replicated i.e. replication operation. *Second*, group convolutions has to be followed by $1 \times 1$ conv to avoid loss of accuracy due to lack of inter-group communication (Zhang et al., 2018), increasing network depth, however CoMNet is free from such constraint.

## 4.2 COLUMNAR STRUCTURE

Our second contribution is to bring columnar organization into `ACM`. In a cortical column, several neurons are connected both parally and serially (Mountcastle, 1997). The parallel connections get realized implicitly via $N$ neurons in a module. However, to realize the serial connection, we stack several `ACMs` one after the another, forming a column, as shown in Figure 2a. Since, `ACMs` are housed in `CCMs`, stacking essentially is performed over `CCMs`. Moreover to respect lateral connection inhibition, an `ACM` of a `CCM` can communicate with exactly one `ACM` of another `CCM`. For example, an `ACM` at $k^{th}$ location in a `CCM` can only feed the `ACM` at $k^{th}$ location in the subsequent `CCM` (Figure 2a).

We also follow the concept of pyramidal neurons which fuses the outputs of several cortical modules. We achieve that by placing a $1 \times 1$ conv ($P_c$) layer after the final `CCM` in the columnar organization. This conv layer fuses the information of multiple modules or `ACMs` by operating at each $(h, w) \in \mathbb{R}^{(M \times N) \times H \times W}$. Each neuron in the $P_c$ consists of a large number of connections arising due to combining the output of $M * N$ channels. It greatly mimics the property of pyramidal neurons. The proposed artificial columnar organization facilitates eliminating $1 \times 1$ layers which results in reduced depth, reduced FLOPs, latency, and faster convergence of CoMNet, as discussed below.

**Reduced Depth, FLOPs, and Latency** As mentioned in Sec. 2, $1 \times 1$ layers are the major constituent of depth in state-of-art networks because they are based on unit/blocks which is a stack of $1 \times 1$, $3 \times 3$, and $1 \times 1$ layers. Several such units are connected serially to form a stage. For instance, in a stage with three units, six $1 \times 1$ layers are present for three $3 \times 3$ layers. Since, receptive field is mainly governed by $k \times k$ conv layers with $k > 1$ (Luo et al., 2016), the columnar organization facilitates elimination of $1 \times 1$ layers by stacking $3 \times 3$ `CCM` layers, achieving equivalent receptive

Figure 2: (a) Columnar organization, (b) long range connections (LRC), and (c) CoMNet unit / stage.

in just three layers. In other words, columnar organization reduces the three units into one. Notice that, It is true for any number of units. Interestingly, for this reason, a CoMNet unit acts as a stage in contrast to earlier networks where a unit serves as a block, and multiple of such blocks forms stage which are sequentially connected to build a network. The elimination of $1 \times 1$ results in decreased depth which results in reduced FLOPs and latency because of the reason mentioned in Sec 2. As an example, CoMNet performs better than ResNet-50 at $50\%$ fewer layers, while having lower parameters, FLOPs and latency, indicating a huge achievement.

**Faster Convergence**    In just half epochs i.e. 60, CoMNet achieves $99.17\%$ ($76.16\%$) of its accuracy obtained at 120 epochs ($76.76\%$), as compared to ResNet-50 which achieves only $97\%$ ($74.15\%$ vs $76.32\%$). Primary reason behind these improvements is our belief that the $3 \times 3$ layers are more important than $1 \times 1$ because they are solely responsible for extracting spatial context, thereby governing receptive field. Hence, we can obtain equivalent receptive field without $1 \times 1$ by keeping the number of $3 \times 3$ layers same. However, $3 \times 3$ suffers with overly large parameters for which $1 \times 1$ convolutions were introduced (He et al., 2016). Our CoMNet tackles this issue by employing ACMs.

### 4.3    ARTIFICAL LONG RANGE CONNECTIONS

Our third contribution is to develop artificial long range connections. In a visual cortex, these are realized via pyramidal neurons which apart from communicating with the modules, also projects their input to many layers (Mountcastle, 1997). This helps in taking advantage of multi-layer information.

We achieve the same by employing $1 \times 1$ conv ($L_c$) layer which is fed by pooling the input of the CoMNet unit (discussed next). $L_c$ projects this input directly toward the output of the unit while fusing it with the output of $P_c$. This simulates the behavior of combining the cortical module information and multi-layer information. There are many CCM layers between the input and output of a CoMNet unit, however, $L_c$ bypasses all of them, and directly connect input and output, thereby realizing the effect of long range connections. Although they improve accuracy significantly, they can be traded for parameters, FLOPs at the cost of reduced accuracy. See Figure 2b for visualization.

The long range connections find their structural similarity with projections in (He et al., 2016), however there are notable differences. First, projections in (He et al., 2016) are used only in the first block of a stage, while projection between stages does not exist. Second, projection conv operates with a stride of 2 in (He et al., 2016), whereas in our case, $L_c$ is preceded by a pooling operation. Pooling serves two objectives: First, it helps gathering spatial context because $L_c$ is a pointwise convolution, and aligns the spatial size of the input with output of the final CCM.

### 4.4    THE COMNET UNIT.

The above proposed CoMNet design ideas are combined to develop the fundamental computational unit of CoMNet, as shown in Figure 2c. In this unit, an input tensor $T_i$ is first passed through a $1 \times 1$ conv ($S$) which squeezes the number of channels by a factor $\mathcal{R}$. Its output $T_s$ is then passed to CM_Feeder which is followed by a stack of CCMs. Each CCM is connected with the previous via residual connections. The output of final CMM is fed to the $P_c$ which fuses the output of all ACMs in the final CCM. Now, $T_i$ is fed to $L_c$ after pooling ($p$), which projects the information from previous stage i.e. $T_i$ to the $P_c$ layer. The output of $P_c$ and $L_c$ are summed to produce an output tensor $T_o$. Mathematically, the whole computational unit can be written as follows:

$$T_s = S(T_i), \quad T_{cm} = \text{CM\_Feeder}(T_s), \quad T_{ccm} = \text{CCM}(.) \tag{1}$$

$$T_c = \text{CCM}_l \odot ...\text{CCM}_1 \odot \text{CCM}_0(T_{cm}) \implies \bigodot_{s=0...l} \text{CCM}_s(X_s), X_0 = T_{cm}, X_{s>0} = T_{ccm_{s-1}} \tag{2}$$

$$T_o = P_c(T_c) + L_c(p(T_i)) \tag{3}$$

where, $\odot$ represents function-of-function operator. It can be seen that CoMNet is uni-branched. Although CoMNets are faster even with branch, but it can be eliminated using structural reparameterization (Ding et al., 2021) during inference, if desired. Fewer branching reduce the memory consumption and memory access cost regardless of training or testing. As CoMNet unit is mostly $3 \times 3$ convs, it is fully compatible with hardware accelerators because they have dedicated support for them $3 \times 3$. So far, we have shed light on all of the objectives which we envisaged to achieve.

Table 1: Effect of ACM, Varying $M, N, l$. Values of $M, N, l$ are for each of the four CoMNet stages.

| Row | N | | | | l | | | | M | | | | #Depth | LRC | Residual CCM | #Params | #FLOPs | Top-1 (%) |
|---|---|---|---|---|---|---|---|---|---|---|---|---|---|---|---|---|---|---|
| R0 | 16 | 32 | 64 | 128 | 3 | 4 | 6 | 3 | 1 | 1 | 1 | 1 | 26 | ✓ | ✓ | 8.80M | 1.25B | 74.45 |
| R1 | 16 | 32 | 64 | 128 | 3 | 4 | 6 | 3 | 4 | 4 | 4 | 4 | 26 | ✓ | ✓ | 12.1M | 1.77B | 75.65 |
| R2 | 16 | 32 | 64 | 128 | 3 | 4 | 6 | 3 | 5 | 5 | 5 | 5 | 26 | ✓ | ✓ | 13.2M | 1.95B | 76.01 |
| R3 | 32 | 64 | 128 | 256 | 3 | 4 | 6 | 3 | 1 | 1 | 1 | 1 | 26 | ✓ | ✓ | 11.3M | 1.65B | 75.37 |
| R4 | 32 | 64 | 128 | 256 | 3 | 4 | 6 | 3 | 4 | 4 | 4 | 4 | 26 | ✓ | ✓ | 19.8M | 3.05B | 76.76 |
| R5 | 32 | 64 | 128 | 256 | 3 | 4 | 6 | 3 | 5 | 5 | 5 | 5 | 26 | ✓ | ✓ | 22.6M | 3.51B | 77.01 |
| R6 | 32 | 64 | 128 | 256 | 3 | 4 | 6 | 3 | 1 | 1 | 1 | 1 | 26 | ✓ | ✗ | 11.3M | 1.65B | 75.28 |
| R7 | 32 | 64 | 128 | 256 | 3 | 4 | 6 | 3 | 1 | 1 | 1 | 1 | 26 | ✓ | ✓ | 11.3M | 1.65B | 75.37 |
| R8 | 32 | 64 | 128 | 256 | 4 | 5 | 20 | 3 | 1 | 1 | 1 | 1 | 44 | ✓ | ✗ | 13.4M | 2.12B | 75.18 |
| R9 | 32 | 64 | 128 | 256 | 4 | 5 | 20 | 3 | 1 | 1 | 1 | 1 | 44 | ✓ | ✓ | 13.4M | 2.12B | 75.88 |

## 4.5 COMNET INSTANTIATION

A CoMNet variant can be instantiated by connecting many CoMNet units serially which also serve as stages. Without complicating, we follow earlier models (He et al., 2016; Simonyan & Zisserman, 2014) to keep the tradition of five stages among which first is a plain $3 \times 3$ convolution with stride 2, while remaining are the CoMNet units. Following (He et al., 2016), the channel dimension of $S$ starts with 64 which gets multiplied by 2 at each stage, while channel dimension of $P_c$ and $L_c$ always equals to $\mathcal{R}$ times of the channels in $S$. We set $\mathcal{R} = 4$, following (He et al., 2016). To further simplify the instantiation process, the number of CCM layers i.e. $l$ in $k^{th}$ CoMNet unit is set equal to the number of blocks in the $k^{th}$ stage of ResNet-50 (He et al., 2016). A CoMNet unit mainly has three hyperparameters: $M, N, l$ In this work, we do not explore every permutation and combination of them, instead, the presented CoMNet models in this paper, are derived such that, they can outperform most useful and deployed state-of-the-art models.

## 5 EXPERIMENTS

We present in-silico testing of CoMNet on ImageNet (Deng et al., 2009) which is de-facto standard to benchmark novel architectures. We begin with ablations where we specifically study the significance of ACMs, variation in $M, N, l$, effect of long range connection, and residual connections in CCM. We train each CoMNet variant for 120 epochs using SGD, Nesterov momentum, base_lr=0.1 with cosine-annealing (Loshchilov & Hutter, 2016). We also study 200 epochs regime to demonstrate efficacy of CoMNet against state-of-the-art CNNs which are evaluated for 200 epcohs. For 120 epochs, we use RandomResized crop (Paszke et al., 2019), random flip only, while for 200 epochs, MixUp (Zhang et al., 2017), AutoAugment(Cubuk et al., 2019), LabelSmoothing (Szegedy et al., 2016) are also employed. We also show faster convergence by training CoMNet for 60 epochs.

### 5.1 ABLATION STUDY

**Variation in $M, N$**    Table 1 demonstrate the effect of varying $N$ and $M$ (R0-R5). We first fix the values of $N$ and vary $M$ (R0-R5), and then vary $M$ while fixing $N$ (R0 ↔ R3, R1 ↔ R4, R2 ↔ R5). It can be seen that for fixed $N$, accuracy improves by increasing $M$, and the same effect is seen by fixing $M$ and varying $N$. The purpose of this ablation is to show how accuracy, parameters and FLOPs changes with these hyper-parameters. It can be noticed that parameters, FLOPs can be precisely controlled by changing the $M$ (R1 ↔ R2, R4 ↔ R5) which directly reflects accuracy.

**Effect of ACM**    To analyze the significance of ACM, we compare instances having different $N, M$, but have similar parameters and FLOPs budget, for instance, R1 ↔ R2, R1 ↔ R3, Table 1. It is noticeable that, $R2$ with 5 ACM is better by $0.36\%$ in accuracy, only at $1.1$M more parameters relative to R1. Similarly, $R1$ is better by $0.28\%$ in accuracy, only at $0.8$M more parameters relative to R3. It shows that multiple ACMs facilitates improved accuracy in just a fraction of parameters, and FLOPs. Moreover, if comparing R9 (a deeper model) with R2, we see that R2 with multiple cortical modules achieves $0.13\%$ more accuracy in $0.2$M fewer parameters, and $0.17$B fewer FLOPs. It shows the advantage of having multiple cortical modules while being shallower.

**Varying "$l$"**    The impact of varying $l$ is shown in R9, Table 1. It can be seen that accuracy definitely improves relative to shallower, but depth increases simultaneously. The increased depth causes increased latency, therefore, we stick to $20 - 40$ layers of depth. We also carry out additional experiment where each CCM is followed by a $1 \times 1$ as done in group convolutions, while keeping depth and parameters constant. We observe a drop in accuracy $1\%$.

**Residuals in CCM**    R6-R9, Table 1 shows this analysis for a shallower and a deeper variant. For shallower model, residual connection shows only minor improvement ($0.09\%$) however for deeper model the effect of residual connections is significant ($0.7\%$).

Table 2: Effect of long range connections LRC module.

| Row | N | | | | l | | | | M | | | | #Depth | LRC | Residual CCM | #Params | #FLOPs | Top-1 (%) |
|---|---|---|---|---|---|---|---|---|---|---|---|---|---|---|---|---|---|---|
| R0 | 32 | 64 | 128 | 256 | 3 | 4 | 6 | 3 | 1 | 1 | 1 | 1 | 26 | ✗ | ✓ | 8.5M | 1.29B | 73.61 |
| R1 | 32 | 64 | 128 | 256 | 3 | 4 | 6 | 3 | 1 | 1 | 1 | 1 | 26 | w/o. Pool | ✓ | 9.8M | 1.44B | 74.15 |
| R2 | 32 | 64 | 128 | 256 | 3 | 4 | 6 | 3 | 1 | 1 | 1 | 1 | 26 | w. Pool | ✓ | 9.8M | 1.44B | 75.37 |

Table 3: Minimal design space of CoMNet.

| Model | $P_c$ | | | | l | | | | l | | | | M | | | | #Depth | #Params | #FLOPs | Latency | #Epochs | Top-1 (%) |
|---|---|---|---|---|---|---|---|---|---|---|---|---|---|---|---|---|---|---|---|---|---|---|
| CoMNet-A0 | 256 | 512 | 1024 | 2048 | 16 | 32 | 64 | 128 | 3 | 4 | 6 | 3 | 1 | 1 | 1 | 1 | 26 | 8.8M | 1.25B | 4ms | 120 | 74.45 |
| CoMNet-A1 | 256 | 512 | 1024 | 2048 | 16 | 32 | 64 | 128 | 3 | 4 | 6 | 3 | 4 | 4 | 4 | 4 | 26 | 12.1M | 1.77B | 5ms | 120 | 75.65 |
| CoMNet-A2 | 256 | 512 | 1024 | 2048 | 16 | 32 | 64 | 128 | 3 | 4 | 6 | 3 | 5 | 5 | 5 | 5 | 26 | 13.2M | 1.95B | 7ms | 120 | 76.01 |
| CoMNet-B0 | 256 | 512 | 1024 | 2048 | 32 | 64 | 128 | 256 | 3 | 4 | 6 | 3 | 1 | 1 | 1 | 1 | 26 | 11.3M | 1.65B | 4ms | 120 | 75.37 |
| CoMNet-B1 | 256 | 512 | 1024 | 2048 | 32 | 64 | 128 | 256 | 3 | 4 | 6 | 3 | 4 | 4 | 4 | 4 | 26 | 19.8M | 3.05B | 6ms | 120 | 76.76 |
| CoMNet-B2 | 256 | 512 | 1024 | 2048 | 32 | 64 | 128 | 256 | 3 | 4 | 12 | 3 | 4 | 4 | 4 | 4 | 32 | 23.3M | 3.74B | 8ms | 120 | 77.15 |
| CoMNet-B3 | 256 | 512 | 1024 | 2048 | 32 | 64 | 128 | 256 | 3 | 4 | 6 | 3 | 5 | 5 | 5 | 5 | 32 | 22.6M | 3.51B | 6ms | 120 | 77.01 |
| CoMNet-C0 | 256 | 512 | 1024 | 2048 | 48 | 80 | 144 | 272 | 3 | 4 | 6 | 3 | 4 | 4 | 4 | 4 | 26 | 21.6M | 3.73B | 6ms | 120 | 76.93 |
| CoMNet-C1 | 256 | 512 | 1024 | 2048 | 48 | 80 | 144 | 272 | 4 | 4 | 6 | 4 | 4 | 4 | 4 | 4 | 28 | 24.4M | 4.12B | 7ms | 120 | 77.34 |
| CoMNet-C1 | 256 | 512 | 1024 | 2048 | 48 | 80 | 144 | 272 | 4 | 4 | 6 | 4 | 4 | 4 | 4 | 4 | 28 | 24.4M | 4.12B | 7ms | 200 | 78.54 |
| CoMNet-C2 | 256 | 512 | 1024 | 2048 | 48 | 80 | 144 | 272 | 3 | 4 | 6 | 3 | 6 | 6 | 16 | 6 | 26 | 38.9M | 7.09B | 9ms | 120 | 78.05 |
| CoMNet-C2 | 256 | 512 | 1024 | 2048 | 48 | 80 | 144 | 272 | 3 | 4 | 6 | 3 | 6 | 6 | 16 | 6 | 26 | 38.9M | 7.09B | 9ms | 200 | 79.25 |
| CoMNet-D0 | 256 | 512 | 1024 | 2048 | 64 | 96 | 160 | 288 | 3 | 4 | 6 | 3 | 4 | 4 | 4 | 4 | 26 | 23.6M | 4.52B | 8ms | 120 | 77.25 |
| CoMNet-D1 | 256 | 512 | 1024 | 2048 | 64 | 96 | 160 | 288 | 4 | 5 | 12 | 5 | 4 | 4 | 12 | 4 | 36 | 57.0M | 10.8B | 11ms | 200 | 80.53 |
| CoMNet-E0 | 256 | 512 | 1024 | 2048 | 64 | 128 | 256 | 512 | 3 | 4 | 6 | 3 | 1 | 1 | 1 | 1 | 26 | 18.2M | 2.81B | 6ms | 120 | 76.51 |

**Effect of Long Range Connections (LRC)** As mentioned previously, LRC module can be traded for parameters and FLOPs at the cost of reduced accuracy. However, it is important to analyze its significance. To achieve that, we train a CoMNet instance in three ways: *First*, remove LRC entirely, *Second*, use LRC without pool, and *Third*, use LRC with pool. See Table 2 for the analysis. It can be noticed that without LRC (R0), the model suffers with heavy accuracy loss of $\sim 0.54\%$ relative to when LRC is used without pool (R1). Moreover, when using LRC with pool, accuracy improves significantly i.e. 1.22% and 1.76% relative to R1, and R0 respectively. It happens because pooling operation can provide more spatial context to the $1 \times 1$ $L_c$ layer by summarizing the neighborhood.

### 5.1.1 COMNET DESIGN SPACE

Training many instances by varing hyperparameters of a network is a very time consuming process which requires a lot of computing resources and months of duration, also visible in (Schneider et al., 2017). Even the earlier (He et al., 2016; Simonyan & Zisserman, 2014) and newer ones (Ding et al., 2021) avoids exploring the whole design space due to this reason. ResNet being half a decade older, its design space has only recently been explored (Schneider et al., 2017).

For this reason, despite having only three hyperparameters, we could train only a few CoMNet models that shows superiority over the existing models. However, the accomplishments of CoMNet design encourages us to push ourself to provide a minimal design space of CoMNet (See Table 3). All the CoMNet instances are diverse, and have their five most crucial attributes reported i.e. parameters, FLOPs, Latency, epochs and accuracy. These instances are crafted such that they can easily outperform the most popular deployable networks, and can be used by the community as pretrained ones. Definitely there are always better ways to train i.e. longer training (300 epochs), more complex augmentation (Yun et al., 2019), (Huang et al., 2016) which can be used to further improve the accuracy of these instances.

### 5.1.2 IMAGENET CLASSIFICATION

Next we put CoMNet against existing and popular models which are quite dominant in the academia, industry and real-time applications. We shed light on the multi-dimensional efficiencies achieved by CoMNet. If efficiency is not possible in all dimensions, CoMNet offers stronger trade-offs relative the rival network. We show that CoMNet is simple during both training and inference while can also take advantage of recent structural reparameterization (Ding et al., 2021) during inference phase.

When comparing CoMNet with existing CNNs, if efficiency in every dimension is not feasible, we prioritize a few dimensions over the others by defining a dimension precedence: Latency = Depth > Branching > FLOPs > Parameters. Although parameters efficiency is important, it can be sacrificed if CoMNet is better in other dimensions, while latency is kept at the highest.

Table 4 shows the comparison of CoMNet with existing models over multi-dimensional metrics. Latency is reported for RTX-2070 GPU which may vary on a different GPU device, therefore the numbers are only for reference. Below we discuss the observations in a structured manner.

Table 4: CoMNet in the wild.

| Row | Architecture | #Depth | #Epochs | #Params | #SR_Params | #FLOPs | #SR_FLOPs | Latency | SR_Latency | Top-1 (%) |
|-----|--------------|--------|---------|---------|------------|--------|-----------|---------|------------|-----------|
| R0 | ResNet-18 (He et al., 2016) | 18 | 120 | 11.6M | — | 1.83B | — | 4ms | — | 71.16 |
| | RepVGG-A0 (Ding et al., 2021) | 22 | 120 | 9.10M | 8.30M | 1.51B | 1.46B | 8ms | 4ms | 72.41 |
| | ResNet-34 (He et al., 2016) | 34 | 120 | 21.7M | — | 3.68B | — | 8ms | — | 74.17 |
| | **CoMNet-A0** | 26 | 120 | 8.8M | 1.25M | 1.25B | 1.25B | 7ms | 5ms | 74.45 |
| R1 | RepVGG-A1 (Ding et al., 2021) | 22 | 120 | 14.0M | 12.7M | 2.63B | 2.36B | 7ms | 5ms | 74.46 |
| | EfficientNet-B0 (Ding et al., 2021) | 49 | 120 | 5.26M | — | 0.40B | — | 8ms | — | 75.11 |
| | RepVGG-B0 (Ding et al., 2021) | 28 | 120 | 15.8M | 14.3M | 3.06B | 3.40B | 7ms | 5ms | 75.14 |
| | **CoMNet-A1** | 26 | 120 | 12.1M | 12.1M | 1.77B | 1.77B | 7ms | 5ms | 75.65 |
| R2 | ResNet-50 (Paszke et al., 2019) | 50 | 120 | 25.5M | — | 4.12B | — | 11ms | — | 76.30 |
| | RepVGG-A2 (Ding et al., 2021) | 22 | 120 | 28.1M | 25.5M | 5.69B | 5.12B | 9ms | 7ms | 76.48 |
| | **CoMNet-B1** | 26 | 120 | 19.8M | 19.8M | 3.05B | 3.05B | 7ms | 6ms | 76.76 |
| R3 | ResNet-101 (He et al., 2016) | 101 | 120 | 44.5M | — | 7.85B | — | 15ms | — | 77.21 |
| | ResNeXt-50 (Xie et al., 2017) | 50 | 120 | 25.1M | — | 4.4B | — | 11ms | — | 77.46 |
| | **CoMNet-C1** | 28 | 120 | 24.4M | 24.4M | 4.12B | 4.12B | 7ms | 6ms | 77.34 |
| R4 | ResNet-152 (He et al., 2016) | 152 | 120 | 60.1M | — | 11.5B | — | 15ms | — | 77.78 |
| | ResNeXt-101 (Xie et al., 2017) | 101 | 120 | 44.1M | — | 8.10B | — | 14ms | — | 78.42 |
| | **CoMNet-C2** | 26 | 120 | 38.9M | 38.9M | 7.09B | 7.09B | 11ms | 9ms | 78.05 |
| R5 | RepVGG-B3 (Ding et al., 2021) | 28 | 200 | 123.0M | 110.9M | 29.1B | 26.2B | 22ms | 17ms | 80.52 |
| | RegNetX-12GF (Xie et al., 2017) | 57 | 200 | 46.0M | — | 12.1B | — | 13ms | — | 80.55 |
| | **CoMNet-D1** | 36 | 200 | 57.0M | 57.0M | 10.8B | 10.8B | 12ms | 11ms | 80.53 |
| R6 | ResNet-50 (Paszke et al., 2019) | 50 | 60 | 25.5M | — | 4.12B | — | 10ms | — | 74.15 |
| | **CoMNet-B1** | 26 | 60 | 19.8M | 19.8M | 3.05B | 3.05B | 7ms | 6ms | 76.16 |

**R0.** CoMNet is 3.29% accurate, has 25% fewer parameters, 31% fewer FLOPs than ResNet-18 while offers similar runtime speed. Although it has 6 more layers, CoMNet offers higher representation power in fewer parameters at equivalent latency. Similarly, in comparison to ResNet-34, it is accurate by 0.28%, has 59% fewer parameters, 66% fewer FLOPs, 23% less deeper while runs faster by 37%. In terms of state-of-the-art RepVGG (Ding et al., 2021) models, CoMNet is more accurate by 2.04% while having marginally fewer FLOPs, parameters and equivalent speed.

**R1.** CoMNet is a strong competitor to RepVGG i.e. $\sim 13\%$ fewer parameters, 32% fewer FLOPs while offering similar speeds but higher accuracy by 1.19% for RepVGG-A1. Although we do not target mobile regime in this paper, we still show that having fewer parameters and FLOPs do not guarantee faster speeds. EfficientNet-B0 has 50% fewer parameters and 77% fewer FLOPs, but is 50% more deeper, and runs 37% slower. We believe that by exploring the design space of CoMNet for mobile regime, CoMNet can offer trade-offs to EfficientNet, which we leave for future work.

**R2.** ResNet-50 is the most widely employed backbone for many areas such as (Ren et al., 2015; He et al., 2017; Goyal et al., 2017) due to its affordability in terms of representation power, FLOPs, depth and accuracy. CoMNet easily surpasses ResNet-50 while being 50% shallower, 22% fewer parameters, 25% fewer FLOPs and 40% faster. CoMNet also outperforms RepVGG-A2.

**R3-R4.** The corresponding experiments shows that CoMNet is better than two larger variants of ResNet which despite being older, still serves as backbones for many cutting-edge works (Carion et al., 2020; Li et al., 2022). Our CoMNet outperforms these two baselines in every aspect while surprisingly being 72% and 82% less deeper relative to ResNet-101 and ResNet-152 respectively. CoMNet also runs significantly faster by almost 50% in roughly 50% fewer parameters and FLOPs. This is quite promising in short training epochs while being shallower.

From the table, it can be also be seen that CoMNet despite being smaller then ResNeXt, outperforms in all the dimensions with marginal accuracy difference, which can be improved by matching the parameters of CoMNet with ResNeXt. Overall CoMNet is 50% less deeper then ResNeXt-50 while runs 50% faster at 6% fewer FLOPs, 2% fewer parameters and roughly same accuracy which is huge. Moreover, CoMNet is also 75% less deeper , 11% less parameters, 12% fewer FLOPs and 35% faster at a marginal accuracy difference which can be matched by changing the CoMNet hyperparameters.

**R5.** CoMNet is as accurate as RegNetX-12GF (Schneider et al., 2017) at 10% fewer FLOPs, 36% less depth, and runs 15% faster while offering competitive tradeoffs in the parameters. For such a shallow model, achieving beyond 80% with merely 36 layers in multi-dimensional efficiency setting has not been seen previously. Even RepVGG is dramatically less efficient i.e. 53% fewer parameters, 62% fewer FLOPs while running significantly faster. This shows promising nature of CoMNet in large network regime. However, our target in this paper is the real-time domain only.

**R6.** This experiment shows the faster convergence of CoMNet in half of the original training epochs i.e. 60. It can be seen that CoMNet attains 99.17% (76.16%) of its accuracy obtained at 120 epochs (76.76%), as compared to ResNet-50 which achieves only 97% (74.15% vs 76.32%).

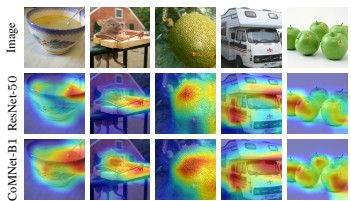

Figure 3: GradCAM visualizations

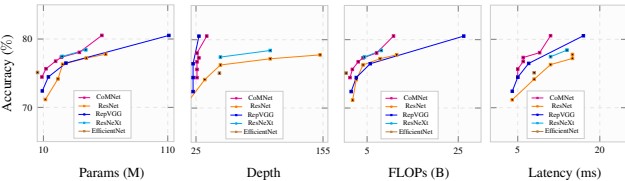

Figure 4: Multi-dimensional efficiency plots.

**Structural Reparameterization.** RepVGG uses structural reparameterization (SR) during inference phase which converts it into a VGG-like (Simonyan & Zisserman, 2014) structure. CoMNet is also compatible with SR and can achieve lower latency but marginally. Interestingly, RepVGG offers lower architectural complexity only during inference phase, but training-phase complexity is still very high i.e. very larger parameters, three branches at each layer (Ding et al., 2021) which leads to increased training time (Table 4). Whereas CoMNet has seamlessly lower complexity during both training and testing, because CoMNet have only one residual connection in CCM layers which can be eliminated via SR. This leads to reduction in FLOPs of element-wise summation layers which is only marginal, however, memory access cost gets reduced due to the branch elimination (Ding et al., 2021). It must be noticed that CoMNet are faster even without SR.

**Multi-dimensional Efficiency.** Figure 4 shows different plots for accuracy vs an objective dimension. It can noticed that CoMNet is always better regardless of the objective, except in depth relative to RepVGG, which is only a difference of $3-4$ layers. Despite that, CoMNet is accurate, has lower latency, lower FLOPs while has low architectural complexity, justifying the objective of this paper.

## 5.2 ADDITIONAL QUALITATIVE EVALUATIONS

**GradCAM: Gradient based Class Activation Map.** To comprehend why CoMNet performs better in multi-dimension, we investigate the class activation maps on ImageNet (Deng et al., 2009) validation set. We use GradCAM (Selvaraju et al., 2017) for this purpose, which computes class activation maps for a class label, indicating the attended regions by the network. GradCAM visualizations of ResNet-50 and CoMNet-B1 (based on R2, Table 4 are shown in Figure 3). It can be seen that CoMNet significantly improves the attended regions of target class relative to the baseline. It indicates high generalizability of CoMNet by emphasizing on class specific parts in the image.

**BrainScore.** Although CoMNet follows the structure of biological visual cortex, we neither claim on CoMNet's functional similarity with brain nor do we aim to build models that are more brain-like (Kubilius et al., 2019). However, as the structure of IT cortex is our major inspiration, we ran BrainScore test on CoMNet-B1 out of curiosity. The BrainScore (Schrimpf et al., 2020) provides a degree to which the intermediate layers of a network behave similar to the layers of a visual cortex. For more details, please refer to (Schrimpf et al., 2020). The BrainScore performance largely depends on the training method. However, in our test, we found that CoMNet ranks 39 (in more than 350 entries) in the IT score without any specialized training. The models ResNet-101 and ResNet-152 are ranked 119 and 143 respectively in the IT score. Notice that our BrainScore analysis merely provides a new perspective on CoMNet, not to assert the functional similarity of CoMNet with visual cortex.

## 5.3 FUTURE DIRECTIONS

Achieving multi-dimensional efficiency is crucial for real-time applications and CoMNet is the first effort in this direction by incorporating biological underpinnings. Despite the significant achievements, CoMNet is open for improvement. *First*, a comprehensive design space of CoMNet can be explored, similar to (Schneider et al., 2017). *Second*, mobile regime of CoMNet is yet to be explored. *Third*, BrainScore can serve as an additional efficiency dimension to improve CoMNet's brain-like behavior.

## 6 CONCLUSION

In this work, we propose a notion of multidimensional efficiency in CNNs, and a novel CNN architecture "CoMNet" to achieve that. CoMNet inherits certain properties of a biological visual cortex such as cortical modules, columnar organization to achieve multi-dimensional efficiency in parameters, FLOPs, accuracy, latency, training duration while having simple architecture. We provide a minimal design space of CoMNet instead of only a few models. CoMNet outperforms many popular networks such as RepVGG, ResNet, RegNet, ResNeXt, while being shallower, significantly faster, and also offering competitive trade-offs when multi-dimensional efficiency is not possible.

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
