# OpenReview forum: "COMNET : CORTICAL MODULES ARE POWERFUL"
_ICLR.cc/2023/Conference — Submitted to ICLR 2023_

### Official Review · Reviewer_8V6y · 2022-10-24

**Confidence:** 3
**Correctness:** 3
**Technical Novelty And Significance:** 3
**Empirical Novelty And Significance:** 2
**Recommendation:** 5

**Clarity, Quality, Novelty And Reproducibility:**

The paper is well written and has details needed to reproduce it. The idea has some novelty inspired by biological networks.

ICLR author guideline states that "Citations within the text should be based on the \texttt{natbib} package
and include the authors' last names and year" which was not followed in this paper.

**Strength And Weaknesses:**

The paper is interesting and provides an alternative architecture that seems to perform well. The paper is well written and should be reproducible from the provided details. The paper has a lot of experiments to study the design of the comnet, however, it is lacking comparisons to other works. There are no comparisons to other network designs, such as group convolutions, or works that explicitly study scaling of networks, e.g., EfficientNet. The claims about controlling parameter growth are a bit misleading, as all the aspects are controllable in other networks, though they might take a bit more work to change all the settings.

**Summary Of The Paper:**

This paper proposes a new CNN architecture inspired by biological networks. Instead of a standard ResNet style architecture, this method duplicates the input over a set of modules that then processes all the inputs. This leads to a different way to control parameters and scale the network, resulting in networks that are less deep, but still perform well. The approach is evaluated on ImageNet.

**Summary Of The Review:**

The paper proposes an interesting idea and has extensive ablation experiments. However, it is lacking comparisons to many existing works, which makes it hard to tell how effective this idea is.

---

> ### Author Response · Authors · 2022-11-16
> **Comparison is based on the settings of RepVGG and most pertinent models are used for comparision. ResNeXt is now added, along with EfficientNet in the multidinesional efficiency plot**
>
> Dear Reviewer, thank you for your valuable feedbacks. We are encouraged that you found our new multidimensional efficient architecture interesting. However, there has been a few concerns of yours which we try our best to address here. However, before that we would like share the contributions in a nutshell for better picture.
>
> we would like to provide a quick recap of the paper:
>
> **Our Goal:**   To bring the notion of multi-dimensional efficiencies in the neural network for real-time applications requiring high representation power. The networks which are easy to customize in terms of latency, depth, parameters, FLOPs etc.
>
> **What is not our goal:** Since CoMNet are the first to focus on multi-dimensional efficiencies, we have focused on design principles while considering in mind to achieve multidimensional efficiencies by taking advantage of the basics of neuroscience. Since, translating biological underpinnings to artificial neural networks itself is challenging, therefore in this paper we focus on presenting the fundamental ideas, and several CoMNet models superior to the popular ones being used in the industry and academia. Hence, we do not focus on theoretical aspects of CoMNet because that is altogether a different field and requires a separate study. We have also mentioned this in the paper (See Sec.1). Moreover, we in this paper are interested in only fully connected convolutions instead of depthwise or mobile series which we target in future work, to develop very small CoMNet models for mobile applications.
>
> Since there are a number of novel concepts in the paper, we would like to compile them in form of contributions in a nutshell.
>
> **Novel Technical Contributions:**
>
> 1. We introduce term of multi-dimensional efficiency which is not available in the literature.
>
> 2. The notion of artificial cortical modules (ACM) which helps achieving high representation in fewer parameters, controlled parameter growth, increased computational density.
>
> 3. The concept of columnar organization which helps to achieve smaller depths, lower latency and FLOPs, faster convergence.
>
> 4.  Long range connections with pooling similar to pyramidal neurons which further improves the accuracy of CoMNet.
>
> **Novel Emperical Contributions:**
>
> 1. Latency is top-priority of CoMNet due to pragmatic utility. Optimizing latency is uncommon in neural network architectures.
>
> 2. We show that it is possible to achieve higher accuracies in smaller depths while having a simple architecture and easy control over network parameters.
>
> 3. We show that our technical contributions lead to faster convergence of CoMNet.
>
> 4. We also provide a detailed ablation study, and a minimal design space of CoMNet which usually is not reported when a new architecture is presented.
>
> 5. CoMNet is superior to a number of recent and older architectures in multiple dimensions.
>
> 6. CoMNet Learns better representation as shown via GradCAM.
>
> 7. CoMNet has higher inferotemporal cortex score than popular and dominent ResNets in BrainScore.
>
> **The paper is interesting and provides an alternative architecture that seems to perform well. The paper is well written and should be reproducible from the provided details. The paper has a lot of experiments to study the design of the comnet**
>
> Thank you for the encouragement
>
> Please read the below response for remaining concerns (provided as a comment to this comment)

---

> > ### Author Response · Authors · 2022-11-16
> > **Contd..**
> >
> > **however, it is lacking comparisons to other works. There are no comparisons to other network designs, such as group convolutions, or works that explicitly study scaling of networks, e.g., EfficientNet.**
> >
> >
> > We respectfully say that our experimental comparision is strongly based on the settings of RepVGG which is a recent popular approach. CoMNet uses group convolutions merely as implementation strategy, not as a fundamental design choice, unlike networks which has groupwise convolution as main engine and fundamental building block such as shufflenetv1. Moreover, grouwise conv is only used in mobileseries networks which are very small. We in this paper only have focused on high representation power networks similar to RepVGG and mobile regime of CoMNet we wish to explore as a separate study.
> >
> >
> > Moreover, EfficientNets are primarily a method of compound scaling and they explicitly study only scaling. Moreover, if we closely look at them, they are mainly MobileNet-v2 but with SE units and compound scaling. Most importantly the resolution scaling which is important.  The reason why we included only EfficientNet-B0 is that EfficientNet are scaled using compound scaling in which resolution is also scaled, and it is well shown that high resolution leads to better accuracy. However, we have followed the standard practice to train on and EfficeintNet-B0 is also trained at 224x224 which is also only the comparison point in RepVGG. EfficientNet-B0 can be found in the Table-2.
> >
> > However upon your feedback, we have also included in the multidimensional efficiency plots. As mentioned in Sec 2, EfficientNet suffer from extreme complexity, hindering their real-time performance despite being efficient in parameters, FLOPs. EfficientNet are not ideal for real-time GPU applications as mentioned in RepVGG paper, because of their high memory access cost and branching.
> >
> > Overall, we believe that we have ample experimentation and most suitable comparison with the recent works, (also acknowledged by you) as well as the most preferred networks in the industry for real-time applications.
> >
> > **Citations**
> >
> > Updated
> >
> > Dear reviewer, hopefully we have answered all of your questions. Kindly let us know specifically, what would you like to see more in the papers in terms of comparison. Looking forward to have a discussion with you, as your feedback are crucial in improving the paper's quality.

---

> ### Author Response · Authors · 2022-11-18
> **New draft of the paper uploaded**
>
> Dear reviewer,
>
> We have updated the draft. Due to change in the citation format, we had to make some space and rewrite several lines.
> Particularly the changes are mentioned below:
>
> 1. Introduction edited and now says that paper also introduce the notion of multi-dimensional efficiency.
> 2. Related work now includes ResNeXt.
> 3. Sec 4.1 updated, clearly stating the differences between group conv and ACM.
> 4. CoMNet unit edited with description on the $\odot$ sign.
> 5. Ablation in ``l'' now includes a note on CCM-P-CCM-P.
> 6. Sec 5.1.2 now includes comparision with RexNeXt.
> 7. Multidimensional efficiency plot now includes ResNeXt and EfficientNet.
> 8. Conclusion updated slightly.
> 9. Reference style aligns with ICLR format.
> 10. Supplementary material contains a visualization CoMNet unit with batch norm and ReLU and other connections.
>
> Thank you

---

### Official Review · Reviewer_6Lbu · 2022-10-24

**Confidence:** 4
**Correctness:** 2
**Technical Novelty And Significance:** 2
**Empirical Novelty And Significance:** 2
**Recommendation:** 5

**Clarity, Quality, Novelty And Reproducibility:**

See above.  While the overall description is relatively clear, the following points are unclear to me:

* similarities/differences compared to resnext
* how the "multi-dimensional efficiency" motivation relates uniquely to the architecture developed in this work, beyond standard accuracy vs compute measures


**Strength And Weaknesses:**

While the resulting system appears to achieve good performance compared to recent RepVGG, the modules look fundamentally the same or similar to ResNeXt (https://arxiv.org/pdf/1611.05431.pdf), which also uses the steps (a-e) described above.  It's not clear to me what the differences are, if any, and what makes this system more performant.  The three that I can see as possibilities are: (i) stacking multiple group convs before recombining (I think resnext only uses one); (ii) structural reparameterization (from RepVGG); (iii) availability of improved group convolution implementations in cuda.  How do each of these contribute to performance?  Are there any other differences beyond these three?

The discussion of optimization on multiple efficiency fronts (flops, depth, accuracy, latency, training iters) seems like a bit of a side-track, even though it is mentioned as a main focus.  Descriptions such as in Sec 1, "to the best of our knowledge, achieving multi-dimensional efficiency has not been explored", suggest this paper explores explicit ways to measure combinations of these dimensions, e.g. for use in optimization or model selection, but instead the focus is architecture development with standard performance measures (see comments below).  I think it would make more sense to describe the work from this standpoint.

Overall, this system seems to perform well, and its performance and sizing parameters are well-measured on imagenet.  However, the main components leading to this appear to be group convs (used in ResNeXt) and structural reparameterization (from RepVGG), which are combined to create the CoMNet model.  Without a clearer focus on how this work differs and builds upon each of these, along with investigation on interactions unique to their combination (such as repeated group convs with SR), it's hard to distinguish what might be potentially unique contributions in this work, beyond these two existing works.



More detailed questions and notes:


* Note that the experiment varying "l" doesn't quite address the question of the effects of stacking group convs ("CCMs"), as the recombination ("P_c") is still applied only at the end of the stack.  To see the effect of stacking these before recombining, it would be interesting to look at different combinations of how often the recombination is applied while keeping depth or parameters approximately constant, e.g.:  tile -> CCM -> CCM -> CCM -> P, vs tile -> CCM -> P -> tile -> CCM -> P.  The second has one additional P but one fewer CCM, so similar overall depth.  What is the behavior of this, is there a speed/acc frontier that suggests a best value for number of group conv stacks before recombining and how does it vary with overall depth?

* It would be good to include more comparison systems, including more efficientnet points (only B0 is mentioned in one table), resnext, resnext implemented using the same nvidia group conv library calls used by this system, and for more general comparison, transformer-based ViT variants.

* fig 4:  does the latency plot in this figure use inference-time structural reparameterization?

* sec 3 acm "controlled parameter growth":  "AMCs provide an absolute control over number of synaptic connections":  how do they achieve this exactly, what values are can be changed or varied in order to control the parameter count in this way?

* sec 2 related work:  As mentioned above, I think ResNeXt needs to be compared to in detail.  Also, connectivity matrices controlling the number of synaptic connections are described in LeCun 1998, which could be good to mention as well.

* eq 2: circled dot notation should be explained/defined in the text, this can be used for several ops e.g. concat, elemwise product, tensor product


**Summary Of The Paper:**

This paper presents a CNN architecture based on a module performing the following sequence:

(a)  1x1 dim reduce ("S")
(b)  channel tiling ("CM_Feeder")
(c)  stacked group conv with residuals ("CCM")
(d)  1x1 combine/remap ("P_c")
(e)  skip-conn residual ("LRC")

These steps are motivated by analogy to the columnar organization of the biological visual cortex, particularly ACM/CCM.  After training, structural reparameterization is applied before inference, leading to improved execution speed.  The method is evaluated on imagenet, finding improved speed/accuracy curves compared to ResNet and RepVGG.


**Summary Of The Review:**

Overall, this system seems to perform well, and its performance and sizing parameters are well-measured on imagenet.  However, the main components leading to this appear to be group convs (used in ResNeXt) and structural reparameterization (from RepVGG), which are combined to create the CoMNet model.  Without a clearer focus on how this work differs and builds upon each of these, along with investigation on interactions unique to their combination (such as repeated group convs with SR), it's hard to distinguish what might be potentially unique contributions in this work, beyond these two existing works.

---

> ### Author Response · Authors · 2022-11-16
> **CoMNet are entirely different from ResNeXt , while SR is not the main component of CoMNet**
>
> Dear Reviewer, we highly appreciate your valuable time in reading the paper thoroughly. As per your reviews, the major concern of your has been similarity with ResNeXt and structural reparameterization (SR). Here, we would like to state that there are significant differences which we elaborate below:
>
> 1. First, ResNeXt is the restructurization of the residual block used in ResNet.
>
> 2. The fundamental structure of the residual block and ResNeXt block remains same i.e. $1\times 1$-$3\times 3$-$1\times 1$.
>
> 3. ResNeXt is also based on the block-wise concept employing projections at block level.
>
> 4. Due to same blockwise architecture as of ResNet, ResNeXt also have very high depths leading to higher latency.
>
> 5. It is important to note that a ResNeXt block is repeated multiple times to construct a stage, however a CoMNet block is itself a stage.
>
> 6. Input replication is not present in ResNeXt. Moreover, the $3\times3$ layers divide the input into groups instead of replication. The first squeeze layer ($S$) is a regular convolution and each group of $3\times3$ convolution only process a smaller part of the input. On the otherhand in CoMNet the input is replicated and each group of the CCM layer has access to all the channels of the $S$ layer.
>
> 7. Although the columnar organization appears as simple as stacking, however it is a major source of reduction in overall depth of the network by eliminatation of $1\times1$ layers which are necessary in groupwise convolutions in ResNeXt and other similar networks to combine information from all the groups. Whereas in CoMNet, the input replication provides access of all channels of the input to all of the groups in all the CCM which forms the columnar organization. Due to this reason, CoMNet does not suffer from the drawback of group convolutions as mentioned in shufflenetv1.
>
> 8. Since, CoMNet has only five stages and no concept of blocks, the Long range conncetions are present only per stage and incorporation of information aggregation using pooling is novel as compared to ResNet, ResNeXt or similar architecture.
>
> 9. Overall, a stage with three blocks of ResNeXt will have 9 layers while an equivalent stage of CoMNet will have only 5 layers, yet outperforming ResNeXt.
>
> 10. Upon your request, we have now added ResNeXt to the comparison tables as well as multi-dimensional efficiency plots. And we observe CoMNet outperforming ResNeXt in multiple domains while sacrificing in accuracy slightly, because of significant parameter differences. We your reference we add the same results here.
>
> Network |   Depth |  Epoch | Params | FLOPs | Latency | Top-1 Accuracy
> ---------| --------| --------| --------| --------| --------| ---------|
> ResNeXt-50 |  50 | 120 | 25.1M | 4.4B | 11ms | 77.46
> CoMNet-C1 | **28** | 120 | **24.4M** | **4.12B** | **6ms** | 77.34
> ---------| --------| --------| --------| --------| --------| ---------|
> ResNeXt-101 | 101 | 120 | 44.1M | 8.1B | 14ms | 78.42
> CoMNet-C2 | **26** | 120 | **38.9M** | **7.09B** | **9ms** | 78.05
>
> From the above numbers, It can be seen that CoMNet despite being smaller then ResNeXt outperforms in all the dimensions except marginal accuracy difference, which can be improved by matching the parameters of CoMNet with ResNeXt. Overall CoMNet is 50% less deeper then ResNeXt-50 while runs 50% faster at 6% fewer FLOPs, 2% fewer parameters and roughly same accuracy which is huge. Moreover, CoMNet is also 75% less deeper , 11% less parameters,  12% fewer FLOPs and 35% faster at a marginal difference of accuracy which can be matched by changing the CoMNet hyperparameters. Hence it is evident that CoMNet is substantially different from ResNeXt, primarily visible by the network depth.
>
> **Structural Reparameterization:**
>
> Structural Reparameterization is a recent concept. Our CoMNet's high performance is not due to SR but due to the biological translation of the ideas. SR is used merely to demonstrate that CoMNet does not rule out recent concepts. From Table-4, it can be noticed that CoMNet is faster even without SR. Hence, we mention that, SR is for demonstration purpose only, our main CoMNet design is not based on SR.
>
> **Group Convolutions in CUDA:**
>
> Our fundamental design comes from the biological underpinnings of visual cortex. However to implement them efficiently, we needed group convolutions which are merely an implementation choice. Even without group convolutions, we attain the same accuracy. In the added comparisons, ResNeXt has been run on the same groupwise convolution implementation as of CoMNet.

---

> > ### Author Response · Authors · 2022-11-16
> > **Contd...**
> >
> > **Multiple Efficiency Fronts:**
> >
> > Achieving multidimensional efficiency is our main focus, and we achieve that by developing CoMNet based on the biological understandings. Indeed, as mentioned in the paper, *to the best of our knowledge, multidimensional efficiencies has not been explored previously*. CoMNet is the first to focus on this area.
> >
> > **paper explores explicit ways to measure combinations of these dimensions, e.g. for use in optimization or model selection, but instead the focus is architecture development with standard performance measures (see comments below).**
> >
> >  We say that our focus in achieving multidimensional efficiencies via CoMNet which is significantly different from the existing ones. Our main focus in the paper is not only to develop a new architecture, but to achieve multidimensional efficiencies as well. Upon your feedback, we have modified the text a bit which clearly mentions this case.
> >
> > **Overall, this system seems to perform well, and its performance and sizing parameters are well-measured on imagenet.**
> >
> > Thank you for acknowledging. Indeed this is the case which is a result of our translation of biological theory into artificial neural networks.
> >
> > **However, the main components leading to this appear to be group convs (used in ResNeXt) and structural reparameterization (from RepVGG), which are combined to create the CoMNet model Without a clearer focus on how this work differs and builds upon each of these, along with investigation on interactions unique to their combination (such as repeated group convs with SR), it's hard to distinguish what might be potentially unique contributions in this work, beyond these two existing works.**
> >
> > We respectfully defend by saying that group convolutions and structural reparameterization are not behind the success of CoMNet. As mentioned earlier, group convolutions are merely an implementation strategy. The main components of CoMNet are are artificial cortical modules, columnar organization and long range connection, however to implement them in form of artificial neural networks, we utilize the existing terminologies such as residual learning, as mentioned in Sec.1. We use group convolution as the implementation choice. Even without group convolutions, there is no difference in the performance except runtime because without grouped convolution implementation, the GPUs are utilized inefficiently, as stated in ResNeXt.
> >
> > Regarding SR,  it is merely to demonstrate the flexibility of CoMNet to recent ideas but CoMNet's large performance and superiority in multiple dimensions is not because of SR, which is also clear from Table-4 which indicates that CoMNets are faster even without SR. Hence, to summarize we have highlighted our primary contributions as below:
> >
> > **Novel Technical Contributions:**
> >
> > 1. We introduce term of multi-dimensional efficiency which is not available in the literature.
> >
> > 2. The notion of artificial cortical modules (ACM) which helps achieving high representation in fewer parameters, controlled parameter growth, increased computational density.
> >
> > 3. The concept of columnar organization which helps to achieve smaller depths, lower latency and FLOPs, faster convergence.
> >
> > 4. Long range connections with pooling similar to pyramidal neurons which further improves the accuracy of CoMNet.
> >
> > **Novel Emperical Contributions:**
> >
> > 1. Latency is top-priority of CoMNet due to pragmatic utility. Optimizing latency is uncommon in neural network architectures.
> >
> > 2. We show that it is possible to achieve higher accuracies in smaller depths while having a simple architecture and easy control over network parameters.
> >
> > 3. We show that our technical contributions lead to faster convergence of CoMNet.
> >
> > 4. We also provide a detailed ablation study, and a minimal design space of CoMNet which usually is not reported when a new architecture is presented.
> >
> > 5. CoMNet is superior to a number of recent and older architectures in multiple dimensions.
> >
> > 6. CoMNet Learns better representation as shown via GradCAM.
> >
> > 7. CoMNet has higher inferotemporal cortex score than popular and dominent ResNets.

---

> > > ### Author Response · Authors · 2022-11-16
> > > **Contdd...**
> > >
> > > **Discussion on Questions:**
> > >
> > > **1. Note that the experiment varying "l" doesn't quite address the question of the effects of stacking group convs ("CCMs"), as the recombination ("P_c") is still applied only at the end of the stack. To see the effect of stacking these before recombining, it would be interesting to look at different combinations of how often the recombination is applied while keeping depth or parameters approximately constant, e.g.: tile -> CCM -> CCM -> CCM -> P, vs tile -> CCM -> P -> tile -> CCM -> P. The second has one additional P but one fewer CCM, so similar overall depth. What is the behavior of this, is there a speed/acc frontier that suggests a best value for number of group conv stacks before recombining and how does it vary with overall depth?**
> > >
> > >
> > > Firstly, as per your feedback, we ran this experiment. However, this structure deviates from the proposed notion of cortical columns, since ACMs are being combined more frequently, which does not resemble a column. We would like to share the observations with you. We ran the experiment of CoMNet-A$1$. While doing so, we encounter that number of $3\times3$ layers becomes half which are mainly responsible for governing receptive field and faster convergence. In all, we observe an accuracy drop of ~1%. As per our columnar hypothesis (Sec 4.2), this happens because of reduction in $3\times3$ layers which are responsible for the receptive field.
> > >
> > >
> > > Network | Params  | Latency | Top-1 Accuracy|
> > > ------| ------| ------| ------|
> > > CoMNet-A1 | 12.1M |  5ms  | 75.65%|
> > > CoMNet-B0 | 11.3M |  4ms  | 75.37%|
> > > ------| ------| ------| ------|
> > > CCM-P-CCM-P |  11.4M | 5ms | 74.88%|
> > >
> > > From the experiment, it is clear that CCM-CCM-CCM-P is better.
> > >
> > > **2. It would be good to include more comparison systems, including more efficientnet points (only B0 is mentioned in one table), resnext, resnext implemented using the same nvidia group conv library calls used by this system, and for more general comparison, transformer-based ViT variants.**
> > >
> > > Our comparison strategy is based on RepVGG. Upon your feedback we have included a comparison for ResNeXt. The reason why we included only EfficientNet-B0 is that EfficientNet are scaled using compund scaling in which resolution is also scaled, and it is well shown that high resolution leads to better accuracy. However, we have followed the standard practice to train on $224 \times 224$ and EfficeintNet-B0 is also trained at 224x224 which is also only the comparison point in RepVGG. While the reason to not compare with transformers is that we have adhered ourself to the real-time deployement for which convnets are still superior.
> > >
> > > **3. fig 4: does the latency plot in this figure use inference-time structural reparameterization?**
> > >
> > >  Yes. The latency is with SR, however CoMNet is still better even without SR.
> > >
> > > **4. sec 3 acm "controlled parameter growth": "AMCs provide an absolute control over number of synaptic connections": how do they achieve this exactly, what values are can be changed or varied in order to control the parameter count in this way?**
> > >
> > > As described in Sec 4.1, each ACM has N neurons. If we vary the values of N, the total number of connections can be changed. To understand that consider two ACMs i.e. ACM0 and ACM1 which are connected sequentially each having N neurons. Now each of the N neurons in ACM1 is connected to each of the N neurons of ACM0. If all the neurons are 3x3, resulting in 9 synaptic connections per neurons, it will result in 9N connections  between ACM0 and AMC1. As each connection in artificial neural networks in represented by a weight or parameter, a total of 9N parameters shall be there between ACM0 and ACM1. Now if we change the the value of N, the total number of parameters can also be changed.
> > >
> > > **5. sec 2 related work: As mentioned above, I think ResNeXt needs to be compared to in detail. Also, connectivity matrices controlling the number of synaptic connections are described in LeCun 1998, which could be good to mention as well.**
> > >
> > > Thank you for the suggestion. We now have included ResNeXt in the related work as well as in the comparison.
> > >
> > > **6. eq 2: circled dot notation should be explained/defined in the text, this can be used for several ops e.g. concat, elemwise product, tensor product**
> > >
> > > Thank you for the suggestion. We have incorporated your feedback in the text.

---

> > > > ### Author Response · Authors · 2022-11-16
> > > > **Contd..**
> > > >
> > > > **Clarity, Quality, Novelty And Reproducibility:**
> > > >
> > > > 1. Please see above for the differences and similarity with ResNeXt. The differences are significant.
> > > >
> > > > 2. CoMNets are designed from a real-time deployment point of view with easier training, testing and customization protocols. Generally, so far in the literature of the CNNs, multidimensional efficiencies have not been considered, mostly, accruacy and then FLOPs are considered, which are the standard measures of compute. However considering the rising demands of neural networks, multidimensional efficiencies are crucial where we have added depth, latency as the priority because smaller runtime is the most desired attribute of neural networks in for real-time deployment while accuracy and parameters efficiency can be sacrificed as per the demands in favor of latency.
> > > >
> > > > In order to achieve that, CoMNet are the first to focus on multi-dimensional efficiencies by taking advantage of the basics of neuroscience. Since, translating biological underpinnings to artificial neural networks itself is challenging, therefore in this paper we focus on presenting the fundamental ideas, and several CoMNet models are superior to the popular ones being used in the industry and academia. The scaling of CoMNets is still pending which can be easily performed using compound scaling methods. However still to our best effort, we have provided many comnet models surpassing the performance of popular existing networks.
> > > >
> > > >
> > > > Dear reviewer, hopefully we have answered all your questions. If you need further clarification, Kindly let us know. looking forward to a discussion.

---

> ### Author Response · Authors · 2022-11-18
> **New draft of the paper uploaded**
>
> Dear reviewer,
>
> We have updated the draft. Due to change in the citation format, we had to make some space and rewrite several lines.
> Particularly the changes are mentioned below:
>
> 1. Introduction edited and now says that paper also introduce the notion of multi-dimensional efficiency.
> 2. Related work now includes ResNeXt.
> 3. Sec 4.1 updated, clearly stating the differences between group conv and ACM.
> 4. CoMNet unit edited with description on the $\odot$ sign.
> 5. Ablation in ``l'' now includes a note on CCM-P-CCM-P.
> 6. Sec 5.1.2 now includes comparision with RexNeXt.
> 7. Multidimensional efficiency plot now includes ResNeXt and EfficientNet.
> 8. Conclusion updated slightly.
> 9. Reference style aligns with ICLR format.
> 10. Supplementary material contains a visualization CoMNet unit with batch norm and ReLU and other connections.
>
> Thank you

---

> ### Author Response · Authors · 2022-12-05
> **equivalence exists but under loose conditions (Moving the comment outside due to a complex hierarchicy of earlier responses and answers))**
>
> Dear reviewer, thanks for responding in details to the our responses. Below we discuss the final concerns of yours.
>
>
> **where $W$ is a conv kernel size , which also can be viewed as matrix of size $MN \times C \times k \times k$. Now, reshape $W$ into $M$ kernels $W_i, i=1...M$ each of shape $N \times C k k$. Then the operation of $W$
> is the same as replicate and group conv. This means that replication followed by a single group conv is the same as a regular conv with stacked weights. So in ComNet, the sequence CM-Feeder -> CCM0 is the same as a convolution. One large caveat, though, is depending on how batchnorm/layernorm is used, that part could be different. Otherwise, I think S -> CM-Feeder -> CCM0 -> CCM1 -> ... is the same as S -> 3x3 Conv -> CCM1 -> ... This is perhaps a relatively minor point, just another way to look at the tiling: it combined with the first group conv, is the same as a regular conv (except for possible differences in normalizations). The stacking of group convs, rather than immediate 1x1 between all channels, is different from ResNeXt, which only applies one set of group convs (and does not have an all-to-all 3x3 between any of them).**
>
> Firstly, yes splitting and slicing may have equivalence ***iff*** we discard the channel counts. Thank you for the insighted review. But respectfully, we say that for a given number of groups ($M$), the difference will be in the number of channels in the weights which is not included in your illustrations. So for slicing the weights shall be $MN \times \frac{C}{M} \times k \times k$ whereas for replication it shall be $MN \times C \times k \times k$.
>
> Secondly, yes, S -> CM-Feeder -> CCM0 -> CCM1 -> ... is the same as S -> 3x3 Conv -> CCM1 -> ... and which is as seen by you is just another way to look at the tiling: it combined with the first group conv, is the same as a regular conv (except for possible differences in normalizations).
>
> However, Our explanation has been aligned with that of biological cortical structure where replication occurs. Mathematically, this operation reduces to a plain conv followed by a multiple parallel ACM which takes advantage of group conv. So yes, your observation is correct. If we remove the CM-feeder from our design, then the motivation of biological visual cortex design will not be fully justified. It is a coincident that this operation can be reduced into a plain conv followed by group conv, which is essentially an implementation choice i.e. either compute each ACM separately, or compute them combined via group-conv routines. However to avoid confusion and based  on your feedback  we add this as a note in the appendi  in a new section called implementation directions.
>
> **The stacking of group convs, rather than immediate 1x1 between all channels, is different from ResNeXt, which only applies one set of group convs (and does not have an all-to-all 3x3 between any of them).**
>
> Indeed, this is the major difference which helps achieving smaller depths. Overall, in our defense we say that it is just another way of interpretation from implementation point of view which is very minor issue, also acknowledged by you.
>
> Please read the below response for remaining concerns (provided as a comment to this comment)

---

> > ### Author Response · Authors · 2022-12-05
> > **Contd...**
> >
> > **I read the responses and revisions now in detail. several of my  questions were addressed. In particular**
> >
> > **1. Additional comparisons with ResNeXt are good. It's interesting that in Fig 4, ResNeXt and ComNet appear similar in terms of FLOPs, but ComNet is much faster in latency wall time.**
> >
> > **2. Thanks for pointing out the timings without SR in Table 4, I agree now much of the improvement appears to be before SR is applied, and another good speed increase comes from adding SR in addition, though it's a little hard to see that in the table. It would be easier if Fig 4 also included latency without SR.**
> >
> > **3. The CCM-P-CCM-P is also interesting to see, as this starts to illustrate the effects of directly stacking more group convs.**
> >
> > Thank you for acknowledging that our responses addressed most of your major concerns.
> >
> > **1. I still don't understand the "multidimensional efficiency". It looks like this means the system is evaluated with multiple measures, i.e. the accuracy vs X plots in Figure 4, for different X (params, latency, etc). But comparisons in Fig 4 are in many works, except perhaps depth, which is not as common as params, flops or latency. So I don't really see what is new about "multidimensional efficiency", unless I'm missing something? If this is all that is intended, though, I think stressing it this much in the paper is a little confusing, detracting from clarity of the presentation.**
> >
> > Dear reviewer, as described before,  we have plotted the multiple dimensions to show that CoMNet is better in many dimensions at once, not in only one, while offering competitive trade-offs. As mentioned in Sec-1, that multi-dimension efficiency is crucial and current need of the time. It is so because they are crucial in many applications such as such as autonomous driving, robotics, computer vision which requires real time inference, high accuracy, lower latency instead of throughput because per-sample inference is desirable in real-time applications instead of multi-frame processing. However, existing designs mainly focus on improving accuracy while taking care only one of the dimensions mainly parameters or FLOPs and throughput. Depth and latency are largely ignored however they are extremely important.
> >
> > Lower latency can be obtained by making a network size smaller (small parameters) however it limits the representation power of a network, thus limits the performance of the the network for a given task. What if we want high representation power to achieve high accuracy on the task while also demand lower latency. In the existing design, going deeper has become customary which increases the latency of the networks despite the network is very small. Hence, keeping high representation power along with lower latency is dramatically challenging then it appears. Having overly large parameters also causes parameter overflow, and thus overfitting. Hence, it is desirable to have a balance and full control over the network dimensions. Moreover  a network has either lower FLOPs and high latency while have higher FLOPs and lower latency in fully connected regime. Whereas CoMNet is efficient in both of them i.e. lower FLOPs and lower latency and this is the biggest achievement of CoMNet.
> >
> > To the best of our knowledge, focusing on these critical issues in presence of each other is not addressed which sets apart this work from others, and the resulting CoMNet addresses the multi-dimensional issue effectively by leveraging biological studies.
> >
> > If we do not present the multi-dimensional efficiency metrics and only report FLOPs and parameters, then there is no point of developing a new architecture as existing ones can be tweaked but yes, at the cost of increased architecture complexity or design hyperparameters while still helplessly sacrificing some design constraints in order to satisfy only a few others.
> >
> > To avoid that, it is necessary to bring in the multidimensional efficiency part into the paper which forms the foundation that CoMNet is not merely yet another CNN but an architecture with specific goals of multi-dimensional in contrast to existing ones which mainly focus on accuracy or FLOPs. Moreover, we don't encounter any CNN architecture which emphasize on latency with full force while also being efficient in other dimensions. The above explanation can be also inferred from the Sec-1, Sec-2 , by reading in continuity.
> >
> > Finally, this term is a small component of the paper and is only present in paper in the introduction, comnet design and comnet comparison, rest it is not stressed out. Even with saying that we are open to your suggestions to modify the writeup.

---

> > > ### Author Response · Authors · 2022-12-08
> > > **Contd..**
> > >
> > > **Table 4 latency plot without SR**
> > >
> > > Added one more plot for the same.
> > >
> > > **As discussed, tiling followed by one group conv is the same as a single regular conv, except for possible batchnorm. This is a relatively minor point on its own......... Still, the performance of the system does suggest that stacking group convs in the way this system does can be effective.**
> > >
> > > Please see the explanation above.
> > >
> > > Dear reviewer, hopefully we resolved your final concerns. We eagerly want to have your valuable feedback on them. Please let us know any further concerns of yours.
> > >
> > > Looking forward to a discussion.

---

### Official Review · Reviewer_fyC8 · 2022-10-24

**Confidence:** 5
**Correctness:** 3
**Technical Novelty And Significance:** 3
**Empirical Novelty And Significance:** 3
**Recommendation:** 6

**Clarity, Quality, Novelty And Reproducibility:**

This is a very interesting paper and I found the contributions of this work to be novel.

**Strength And Weaknesses:**

The paper has following strengths:

1. I found the idea in this paper to be very interesting but I am not knowledgeable about neuroscience side of things so I cannot evaluate the accuracy of authors’ design decisions from neuroscience point of view.

2. The approach is well-explained, and it was still interesting to read about the relationships with biological cortex.

3. Results clearly demonstrate the superiority over RepVGG and ResNets.

4. Good ablations are conducted and analysis using GradCAM, etc., is conducted to understand the feature representation power of these new networks.


The paper has following weaknesses:

1. I was hoping to see some more theoretical (or at least empirical) study on *why* the new proposed architecture would result in better representative power than traditional blocks of the same size. For example, why would the new way of grouping (e.g., after replicating the input features) should do better (in representative power) than other alternatives (e.g., standard group convolutions when scaled up to match params/flops of CoMNet units)? Since the proposed method does look similar to Group Convolutions, one important ablation should be to replace CCM with group convolution equivalents and then scaling up the model to match params/FLOPs, and then see if we get equivalent accuracy or not. Indeed, if it could be theoretically established why the new architecture would certainly do better than these alternatives, that would be ideal.

2. No comparison against newer state-of-the-art methods like ConvNexts [https://arxiv.org/abs/2201.03545] or their newer scalings shown in recent works like Restructurable Activation Networks (RANs) [https://arxiv.org/abs/2208.08562] is performed which makes it unclear if these models would actually beat the newer models on ImageNet. In any case, these newer papers should be discussed in related work to reflect the latest advances in model design for image classification.

3. It is quite unclear where in the model the non-linear activation functions (e.g., ReLUs) happen. Were any batchnorms/layernoms used to stabilize training? What exact activation functions were used for CoMNet and other baselines?

4. Minor -- Citation format deviates from ICLR official template.


**Summary Of The Paper:**

The paper presents CoMNets, a biologically inspired deep network architecture that outperforms existing methods like ResNets and RepVGG in multiple respects (e.g., Latency, FLOPs, depth, etc.). Experiments are shown on ImageNet.

**Summary Of The Review:**

My point 1 above in the weaknesses section is most critical. It is important to know that some naïve modification to existing ideas (like group convolutions) will not remove the need to have the CCM modules. Theoretical support would be even better. If this point is reasonably addressed, I will raise my score. Point 2 would help too but it is not as critical as point 1 above.

==== UPDATE AFTER REBUTTAL ====
I have read other reviews and author response. Authors have addressed one of my main concerns on comparison against vanilla group convolutions. Therefore, I have raised the score to 6 (marginally above acceptance threshold). Future improvements on this work can include: (1) better theoretical justifications (I am not an expert in biological explanations that authors provided during the rebuttal), and (2) deployment on constrained devices (current models are suitable for desktop-scale GPUs but it may be interesting to see how these ideas can be scaled down to MobileNet scale networks). These directions can be good future works and are not necessary for the current paper.

---

> ### Author Response · Authors · 2022-11-16
> **Theoretical explanation, New experiment.**
>
> Dear Reviewer, Thank you for reading our paper in depth. We are encouraged that you found it interesting to read the paper's content. Despite the fact, there has been some concerns of yours which we try our best to resolve here. Before doing that, we would like to provide a quick recap of the paper:
>
> **Our Goal:**   To bring the notion of multi-dimensional efficiencies in the neural network for real-time applications requiring high representation power. The networks which are easy to customize in terms of latency, depth, parameters, FLOPs etc.
>
> **What is not our goal:** Since CoMNet are the first to focus on multi-dimensional efficiencies, we have focused on design principles while considering in mind to achieve multidimensional efficiencies by taking advantage of the basics of neuroscience. Since, translating biological underpinnings to artificial neural networks itself is challenging, therefore in this paper we focus on presenting the fundamental ideas, and several CoMNet models superior to the popular ones being used in the industry and academia. Hence, we do not focus on theoretical aspects of CoMNet because that is altogether a different field and requires a separate study. We have also mentioned this in the paper (See Sec.1). Moreover, we in this paper are interested in only fully connected convolutions instead of depthwise or mobile series which we target in future work, to develop very small CoMNet models for mobile applications.
>
> Since there are a number of novel concepts in the paper, we would like to compile them in form of contributions in a nutshell.
>
> **Novel Technical Contributions:**
>
> 1. We introduce term of multi-dimensional efficiency which is not available in the literature.
>
> 2. The notion of artificial cortical modules (ACM) which helps achieving high representation in fewer parameters, controlled parameter growth, increased computational density.
>
> 3. The concept of columnar organization which helps to achieve smaller depths, lower latency and FLOPs, faster convergence.
>
> 4.  Long range connections with pooling similar to pyramidal neurons which further improves the accuracy of CoMNet.
>
> **Novel Emperical Contributions:**
>
> 1. Latency is top-priority of CoMNet due to pragmatic utility. Optimizing latency is uncommon in neural network architectures.
>
> 2. We show that it is possible to achieve higher accuracies in smaller depths while having a simple architecture and easy control over network parameters.
>
> 3. We show that our technical contributions lead to faster convergence of CoMNet.
>
> 4. We also provide a detailed ablation study, and a minimal design space of CoMNet which usually is not reported when a new architecture is presented.
>
> 5. CoMNet is superior to a number of recent and older architectures in multiple dimensions.
>
> 6. CoMNet Learns better representation as shown via GradCAM.
>
> 7. CoMNet has higher inferotemporal cortex score than popular and dominent ResNets in BrainScore.
>
>
> **Addressing Concerns**
>
> **1. I was hoping to see some more theoretical (or at least empirical) study on why the new proposed architecture would result in better representative power than traditional blocks of the same size. For example, why would the new way of grouping (e.g., after replicating the input features) should do better (in representative power) than other alternatives (e.g., standard group convolutions when scaled up to match params/flops of CoMNet units)?**
>
> As we have translated biological underpinnings to CoMNet, we could only provide a few results that it works better than the existing ones and offer an easier control and customization. Better representation in our experience comes from our hypothesis (see Sec 4.1) that *multiple neurons with lesser synaptic connections are better than a single neuron with large connections. For any stimuli, it requires certain amount of representation power or neurons to be learnt. In the existing CNNs, a kernel operates on a large number of channels, thus having large number of synaptic connections. During learning phase, it gets penalized for all the visual stimuli even if it is uncorrelated with them. This is also visible by weight pruning which suggests that post training, eliminating many channels/connections does not impact accuracy, indicating a wastage of synaptic connections.*
>
> We also have shown this empirically in Sec 5.1 under *varying M,N*, and *Effect of ACM*

---

> > ### Author Response · Authors · 2022-11-16
> > **Contd...**
> >
> > **1. contd.. Since the proposed method does look similar to Group Convolutions, one important ablation should be to replace CCM with group convolution equivalents and then scaling up the model to match params/FLOPs, and then see if we get equivalent accuracy or not. Indeed, if it could be theoretically established why the new architecture would certainly do better than these alternatives, that would be ideal.**
> >
> > As mentioned in Sec. 4.1., *Although concept of groups is widely explored in e.g. shufflenet, however there is a key difference. These approaches divide the input channels into groups i.e. a slicing operation. Whereas in CoMNet, input from previous stages in replicated i.e. replication operation, which is fed to multiple ACMs at once. This is the key difference between CoMNet and CNNs employing group convolutions.*' Moreover, these networks have fundamentally similar structure to ResNet style architecture with only differences of groups and depthwise convolutions in shufflenetv2 which is costly in terms of memory access, as compared to full convolutions, as shown in RepVGG paper.
> >
> > Since group convolutions divide the input into groups therefore, they poses an important problem during network customization i.e. the number of input channels must be exactly be divisible by the total number of groups. This appears minor initially, however from implementation point of view, it bounds the user to either increase the number of input channels or decrease, in the multiple of groups. It leads to  non-uniform increment in the parameters. For example,  with $1024$ input channels and $32$ groups group conv works great. However, for arbitrary groups $33$, the user have to change $1024$ to a multiple of $33$ which will either decrease or increase the number of parameters without explicit control. Moreover, in case of transfer learning, if groups are changes, $1024$ channels have to be changed and learning from scratch will have to be carried out. On the other hand CoMNet architecture is extremely easier to customize, possible because of CM\_feeder. Since all the input is replicated, now groups can arbitrarily increased or decreased while keeping $1024$ channels fixed. Interestingly, during trasnfer learning, the unchanged layers need not to learn from scratch.
> >
> > **Experiment**  To further clear your concern, we conducted the experiment mentioned by you and would like to present the analysis.
> >  we removed the CM\_Feeder in all the five stages of CoMNet-A$1$ and used regular group convolutions. The resulting model have parameters similar to CoMNet-A$0$ but performs **2% inferior** while perform **3% inferior** than CoMNet-A$1$. While conducting the experiment, we faced the same problem as mentioned earlier that input channels must be divisible by number of channels. As we increase  the groups to  5 or some other arbitrary value, we are bound to change the dimension of $S$ layer which is a drawback of group convolutions in terms of design and scaling. While in terms of accuracy, group convolutions process only a fewer channels of input which leads to accuracy issues due to low information as mentioned in shufflenet-v1 paper, however CoMNet processes all the input from $S$ but later separates the information into columns.
> > %
> >
> >
> > **2. No comparison against newer state-of-the-art methods like ConvNexts [https://arxiv.org/abs/2201.03545] or their newer scalings shown in recent works like Restructurable Activation Networks (RANs) [https://arxiv.org/abs/2208.08562] is performed which makes it unclear if these models would actually beat the newer models on ImageNet. In any case, these newer papers should be discussed in related work to reflect the latest advances in model design for image classification.**
> >
> > Thank you for the suggestion, we have included these works into related works. However, we would like bring into the note that, ConvNeXt are the restructurization of the traditional residual block of $1\times 1$ - $3\times 3$ - $1\times 1$, to $7 \times 7$ - $1\times 1$ - $1\times 1$ and employ several training tricks, such as high resolution training, pretraining on ILSVRC'22 and then finetuning on ILSVRC'12 which is definitely a reason for improved accruacies. If compare with CoMNet in terms of design, ConvNeXt still have very high depth due to block wise architecture, only focus on accuracy and parameters, and there is no mention of latency which is the most crucial for real-time applications.
> >
> > While the restructurable activation networks provide a stratgy to choose among regular residual blocks and inverted residual blocks, leading to increased depth. On the other hand CoMNet does not have a provision of blocks, since it only has five stage. Hence, design is much more simpler and less deeper which is the result of columnar organization.

---

> > > ### Author Response · Authors · 2022-11-16
> > > **Contd..**
> > >
> > > **3. It is quite unclear where in the model the non-linear activation functions (e.g., ReLUs) happen. Were any batchnorms/layernoms used to stabilize training? What exact activation functions were used for CoMNet and other baselines?**
> > >
> > >  All the convolutions are followed by batchnormalization and ReLU activation. A more detailed network figure will be added to supplementary material.
> > >
> > >
> > > **Citations**
> > >
> > >  Citations corrected.
> > >
> > > Dear reviewer, hopefully, we have addressed all of your concerns respectfully. If you would like to have more details, kindly let us know for further discussion.

---

> ### Author Response · Authors · 2022-11-18
> **New draft updated**
>
> Dear reviewer,
>
> We have updated the draft. Due to change in the citation format, we had to make some space and rewrite several lines.
> Particularly the changes are mentioned below:
>
> 1. Introduction edited and now says that paper also introduce the notion of multi-dimensional efficiency.
> 2. Related work now includes ResNeXt.
> 3. Sec 4.1 updated, clearly stating the differences between group conv and ACM.
> 4. CoMNet unit edited with description on the $\odot$ sign.
> 5. Ablation in ``l'' now includes a note on CCM-P-CCM-P.
> 6. Sec 5.1.2 now includes comparision with RexNeXt.
> 7. Multidimensional efficiency plot now includes ResNeXt and EfficientNet.
> 8. Conclusion updated slightly.
> 9. Reference style aligns with ICLR format.
> 10. Supplementary material contains a visualization CoMNet unit with batch norm and ReLU and other connections.
>
> Thank you

---

### Author Response · Authors · 2022-11-18
**Paper Draft Updated. Kindly acknowledge rebuttal**

Dear reviewers,

As we have updated the paper draft and also posted all the rebuttals, kindly acknowledge a discussion.

Thank you.

---

### Decision · Program_Chairs · 2023-01-20

**Decision:**

Reject

**Justification For Why Not Higher Score:**

More ablations necessary. Minimal empirical gains.

**Justification For Why Not Lower Score:**

n/a

**Metareview: Summary, Strengths And Weaknesses:**

In this work, the authors propose a new variation of convolutional neural networks inspired by biological networks. The method resembles a tiled version of a group convolution operator embedded within a larger organization of interconnected skip connections to resemble the columnar organization of cortical columns. In particular, the authors explore how to employ their architecture in a “plug-in” manner on ResNet, EfficientNet, RepVGG and other meta-architectures for convolutional neural networks. The authors additionally perform some analysis of visualizations through GradCAM as well as a measurement of the BrainScore (a score that reflects the ability of the model to capture the stimulus-response relationship in in-vivo physiological measurements).  The authors argue that their method achieves competitive if not state-of-the-art performance on ImageNet but with a smaller computational demand or a smaller parameter count.

The reviewers commented positively on a novel direction and the empirical success of the results as well as the clarity of the presentation. In particular, the non-trivial gains on ImageNet when applied to established network architectures appeared consistent although admittedly small. The reviewers did express concerns about a few items including (1) a lack of discussion about the theoretical properties about the proposed architecture, (2) the lack of comparison with newer state of the art architectures, and (3) concerns about whether the gains are due to the use of prior methods such as group convolutions and not due to the proposed module itself. During subsequent discussion, one reviewer found the discussion on group convolutions mitigated their concern – in particular due to the follow up baselines studies comparing against an involving group convolutions. However, the discussion with the other reviewer appeared unresolved in spite of a long back-and-forth.

Given that no reviewer was willing to champion the paper for acceptance, I took the opportunity to better understand the issue to see if I could argue in favor of acceptance. Upon my review, I shared the concerns of previous reviewers about the relationship to group convolutions as well as concerns about which specific parts of their multi-part architecture contributed to the performance gains. The subsequent ablations with group convolutions appeared encouraging but given that this issue is of critical importance to the acceptance of the paper, I would have preferred to see these results presented at the time of submission to allow for a more in-depth discussion. For instance, if the end result of the study is to isolate a small modification of group convolutions, then we need to understand what was the critical small adjustment that led to the gain. If indeed the modifications with respect to group convolutions are quite significant, then we need to understand how all of the constituent parts contributed to these gains.

While I applaud the author's efforts attempting to derive meta-architectures from biological insight, I also recognize that this area of endeavor is quite challenging and warrants scrutiny about what exactly we learned from biology with the aim of identifying parsimonious insights. Given these unresolved questions, this paper will not be accepted into this conference but I encourage the authors to follow up on some of these experiments discussed above with a goal of submitting to a future venue at the intersection of machine learning and neuroscience.